# Adsorption of Reactive Blue 116 Dye and Reactive Yellow 81 Dye from Aqueous Solutions by Multi-Walled Carbon Nanotubes

**DOI:** 10.3390/ma13122757

**Published:** 2020-06-18

**Authors:** Christian De Benedetto, Anastasia Macario, Carlo Siciliano, Janos B. Nagy, Pierantonio De Luca

**Affiliations:** 1Dipartimento di Ingegneria per l’Ambiente, University of Calabria, Arcavacata di Rende, I-87036 Arcavacata di Rende, Italy; ch.debenedetto@gmail.com (C.D.B.); anastasia.macario@unical.it (A.M.); 2Dipartimento di Farmacia e Scienze della Salute e della Nutrizione, University of Calabria, I-87036 Arcavacata di Rende, Italy; carlo.siciliano@unical.it; 3Dipartimento di Ingegneria Meccanica, Energetica e Gestionale, University of Calabria, I-87036 Arcavacata di Rende, Italy; janos.bnagy1@gmail.com

**Keywords:** adsorption, carbon nanotubes, non-linear isotherm fitting, Reactive Blue 116, Reactive Yellow 81

## Abstract

The multi-walled carbon nanotubes obtained by catalytic chemical vapour deposition synthesis are used as a solid matrix for the adsorption of the Reactive Blue 116 dye and the Reactive Yellow 81 dye from aqueous solutions at different pH values. The batch tests carried out allowed us to investigate the different effects of pH (2, 4, 7, 9 and 12) and of the contact time (2.5 ÷ 240 min) used. The liquid phase was analysed using ultraviolet-visible spectrophotometry in order to characterise the adsorption kinetics, the transport mechanisms and the adsorption isotherms. The adsorption of the optimal dye was observed at pH 2 and 12. The pseudo-first order kinetic model provided the best approximation of experimental data compared to the pseudo-second order kinetic model. The predominant transport mechanism investigated with the Weber and Morris method was molecular diffusion for both Reactive Yellow 81 and Reactive Blue 116, and the equilibrium data were better adapted to the Langmuir isothermal model. The maximum adsorption capacity for Reactive Yellow 81 and Reactive Blue 116 occurred with values of 33.859 mg g^−1^ and 32.968 mg g^−1^, respectively.

## 1. Introduction

Of all the water volume present on the Earth’s surface, only 0.62% is directly usable by man for vital purposes [1], and in the case of contamination phenomena, the percentage is further reduced. This problem, unfortunately, is easily found in different territorial contexts; for example, textile wastewater is spilled onto the soil without appropriate treatments, causing serious environmental damage that still persists [2]. Thus, the phenomenon of water contamination is an ever-present problem that pushes research to find the most varied solutions. In fact, these problems can only be addressed through the study and enhancement of different sectors that have in common the purpose of eliminating or reducing the spillage of pollutants into the environment. Among the most popular sectors are those aimed at the production of environmentally sustainable materials to reduce the emission of pollutants upstream [3,4,5,6,7] as well as those aimed at the study and use of materials capable of adsorbing [8,9,10,11] or photodegrading [12,13,14,15,16] the pollutants already present in the environment.

Analysing the major consumers of water resources, it is clear that the agricultural consumption exceeds that of other sectors [17]. Nevertheless, the textile manufacturing industry represents the world’s second leading cause of water pollution. Water is used in all production cycles, and the textile waste released causes serious problems related to the considerable scale of the pH, to the high ratios of COD/BOD (biochemical oxygen demand/chemical oxygen demand), to the presence of countless dye substances, to high concentrations of ammonia nitrogen, to the presence of toxic substances and to inhibition of the biological purification process [18]. Among kinds of wastewater from the various processes, tincture waters cause the greatest concern; they are responsible for the coloration of the wastewater, release surfactants, salts, organic process auxiliaries, sulphide, acidity/alkalinity, formaldehyde, significant quantities of metals and suspended solids [19,20]. According to Legislative Decree 152/2006, which is about the discharge limits, textile wastewater has to be treated. Carbon nanotubes are very versatile materials that can be used in different sectors, thanks to their versatility [21,22,23,24], but a particular interest is offered in the field of water purification through the adsorption and abatement of pollutants in the waters [25]. The adsorption process of carbon nanotubes is dependent on the chemical and physical properties of both the contaminants and the nanotubes themselves. Moreover, the environment in which the adsorption process takes place is particularly influential [26]. The knowledge of the phenomena that occur during the adsorption process of carbon nanotubes is an important aspect for the evaluation of the adsorption capacity and for evaluating the transport of contaminants. The dependence of adsorption capacity as a function of the dimensional proportions of carbon nanotubes has been extensively studied [27,28]. The functionalisation of carbon nanotubes is also an important factor that can improve adsorption [28,29]. Some studies have shown an improvement in the removal of oil from water through the use of adsorbents of carbon nanotubes doped with nanoparticles of ferric oxide [30]. The modification of the adsorbent capacities according to the different chemical and physical properties of the contaminants, such as dimensions, hydrophobicity polarity, etc. has been demonstrated [31,32]. Thanks to the high specific surface area and structural typology of carbon nanotubes (CNTs), the adsorption treatment can obtain a rapid and economic reduction of the polluted wastewater [25]. In this work, the phenomenon of adsorption with multi-walled carbon nanotubes (MWCNTs) is studied for the removal of azo dyes, which represent the most abundant class in the textile industry processes [33,34,35,36,37,38,39,40]. Since textile wastewater often has different pH values depending on the fiber coloring process [18], the aim of the present research was to study how the pH of wastewater containing organic dyes can influence the adsorption process promoted by nanotubes. of carbon. In addition, two different azo dyes, Reactive Yellow 81 (RY-81) and Reactive Blue 116 (RB-116), were chosen to study the influence of the size and of the molecular structure on the adsorption mechanism. Although it is known that pollution is an alteration of the environment that is of both natural [41,42,43] and anthropic origin [44,45], the hope is that the latter can be addressed with greater sensitivity through inexpensive and low-cost methods with less environmental impact, as in the case of adsorption of pollutants by carbon nanotubes.

## 2. Materials and Methods

### 2.1. Wastewater Type Solutions

To carry out the experiments, demineralised water was used as the base of the wastewater type solutions in which the Drimarene Turquoise K-2B dye (C.I. Reactive Blue 116; CAS 61969-03-7), formed by copper phthalocyanine (C_32_H_16_CuN_8_) and a sulfonic group (C_9_H_8_ClN_6_O_4_S_2_) with a total molecular weight of 939.88, was dissolved in the first tests. In the second group tests, it dissolved the Procion Yellow H-E3G dye (C.I. Reactive Yellow 81; CAS 59112-78-6), which has the formula C_52_H_34_Cl_2_N_18_Na_6_O_20_S_6_ and a molecular weight of 1632.18. Both dyes were provided by the Italian Coloreria (Henkel, Lomazzo, Italy); they were correctly dissolved in the demineralised water to have a concentration equal to 100 mg L^−1^.

### 2.2. Adsorbent

The nanotubes used in this work were the same as those tested in our previous applications [25]. The used nanotubes have a specific BET area of 108.70 m^2^/g and an average pore width of 103.70 Å. The multi-walled carbon nanotubes were synthesised using the catalytic chemical vapour deposition technique (CCVD), which returned an adsorbent with a purity of 95%, therefore, with a low presence of amorphous. Further information on the nanotubes used can be found in reference [25]. The solid obtained was not subjected to purification or functionalisation treatments, as that would certainly increase the absorption power; nevertheless, at the same time, this increases the production costs. That is why it was introduced as such in batch experiments.

### 2.3. Adsorption Study

The experiments were carried out in a non-continuous state to evaluate the adsorption capacity of the MWCNTs for the dyes RB-116 and RY-81. The experimental methodology for the treatment of the replicated wastewater in the laboratory involved the preparation of a solution inside a graduated chemical flask with demineralised water, in which the designated dye was dissolved to reach a concentration equal to 100 mg L^−1^. In addition to the neutral solution (pH 7), the acid field with the addition of hydrochloric acid (HCl) (pH 9 and pH 12) and the basic field with the addition of sodium hydroxide (NaOH) (pH 2 and pH 4) were studied. Thanks to the addition of these compounds, five different types of wastewater type solutions were studied, with pH values equal to 2, 4, 7, 9 and 12, in order to recreate the real conditions present in the wastewater of the textile industry cycles. During the adsorption process, the system was always subjected to mechanical stirring and the pH was constantly monitored with the use of a pH meter previously calibrated with the appropriate standards. It was kept constant by adding acid or basic solution drops as needed. Buffer solutions were deliberately not used to avoid the presence of other compounds in the system. The pH changes with respect to the initial system, during the adsorption process, were minimal, with a variation of pH = ± 0.1 such as to require sporadic additions of a few drops of diluted acid or base solution (0.01M). Therefore, the variation in Na^+^ and/or Cl^−^ content was considered completely negligible. Moreover, since the volumes of acid or base added to adjust the pH slightly were very small, it can be assumed that the overall ionic strength of the final solution varies by a completely negligible amount compared to that of the initial solution, thus not influencing the process of adsorption. The solutions defined were previously kept under pH and color concentration control in order to verify that the addition of the compounds, for the pH modification, did not change the concentration of dye and to obtain accurate adsorption results. After a first 48-h rest phase and after all of the control operations, the concentration of dye remained unchanged. The solution was then prepared with a pre-set volume of 20 mL inside a graduated glass beaker, together with a mass of 0.06 g of carbon nanotubes. Then, the batch stirring test is started on a magnetic plate (Heidolph MR Hei-Standard) at intervals between 5 ÷ 240 min. Once the stirring time had elapsed, after the adsorption process, the carbon nanotubes were separated from the solution by filtration through a vacuum pump. The filter used was a borosilicate glass filter funnel (Pyrex, porosity 5). At the end of the separation process, the filter remained colourless, allowing for exclusion of an adsorption of the dye by the filter. The solution obtained was clear without any presence of nanotubes. Finally, the residual solution was analysed using the UV-3100PC instrument at room temperature on square section quartz cuvettes with an optical path of 10 mm. The absorbance measurements are referred to a peak with a wavelength equal to 290 nm for the RB-116 and 350 nm for the RY-81. For each UV analysis, “white” reference solutions were used, not containing dye but containing the same quantities of acid or added base. Therefore, the amount of solute absorbed per adsorbent unit *q_e_* represents the adsorption capacity of a given material, and it is expressed with the following Equation (1):(1)$$q_{e}=\frac{\left(C_{0}-C_{e}\right)}{m}\cdot V$$
where *m* is the mass of adsorbent material and *V* is the volume of liquid, while *C_0_* and *C_e_* represent the initial and equilibrium concentration of the species in the solution, respectively. For each system considered, three samples were prepared and analysed. The standard deviation returned values less than 3%. The values shown are averages.

For the calibration curve, to replicate the analysis about the pH (2, 4, 7, 9 and 12), a “blank” with demineralised water corrected with addition of the compound (NaOH, HCl) was used; these values were read with the software combined with the UV-3100PC instrument (Shimadzu, Kyoto, Japan)

The transport, kinetic and equilibrium models were fitted through non-linear analysis using the Golden Software Grapher^®^ 10.1.640 (Golden, CO, USA); this allowed us to evaluate the accurate correlation of the models by the determination coefficient *R^2^* (2), which is explained below:(2)$$R^{2}=\left(\frac{\sum_{i}^{n}{\left(q_{i,exp}-{\bar{q}}_{i,exp}\right)}^{2}-\sum_{i}^{n}{\left(q_{i,exp}-q_{i,model}\right)}^{2}\;}{\sum_{i}^{n}{\left(q_{i,exp}-{\bar{q}}_{i,exp}\right)}^{2}}\right)$$
where $q_{i,exp\;}$ represents the value of *q* measured experimentally; ${\bar{q}}_{i,exp}$ is the average value of *q* measured experimentally; and $q_{i,model}$ represents the value of *q* foreseen by the interpolation model.

For the kinetic study, the pseudo-first order (3) and pseudo-second order (4) model [46] were used. Some studies have shown that, although the pseudo-first and second order equations are widely used for adaptations of adsorption kinetic data, some applications of these equations are not always precise. In fact, when the adaptation of the model is performed using constant *Qe* corresponding to equilibrium sorption, the instantaneous driving force for sorption risks being underestimated [47].

For the transport mechanisms analysis, the model proposed by Weber and Morris (5) [48] was used, while the Freundlich (6) [49] and Langmuir (7) [50] models were used to analyse the adsorption isotherm:(3)$$q_{\left(t\right)}=q_{e}\left(1-e^{-k_{1}t}\right)$$
(4)$$q_{\left(t\right)}=\frac{k_{2}\cdot q_{e}^{2}\cdot t}{1+k_{2}\cdot q_{e}\cdot t}$$
where *q*_(*t*)_ represents the adsorption capacity [mg_dye_ g^−1^_CNTs_]; *q_e_* is the adsorption capacity that is reached at equilibrium [mg_dye_ g^−1^_CNTs_] (therefore it is possible to interpret it as maximum capacity); *k_1_* is the kinetic constant of the pseudo-first order [min^−1^]; *k_2_* is the kinetic constant of the pseudo-second order [min^−1^]; and finally *t* is the contact time [min].
(5)$$q_{\left(t\right)}=k_{id}\cdot \sqrt{t}+C$$
where *k_id_* is the intraparticle diffusion constant expressed in [mg_dye_ g^−1^_CNTs_ min^−1/2^]; *C* is a constant relative to the thickness of the boundary layer expressed in [mg_dye_ g^−1^_CNTs_]:(6)$$q_{e}=K_{F}C_{e}{}^{1/n_{F}}$$
(7)$$q_{e}=\frac{q_{MAX}\cdot K_{L}C_{e}}{1+K_{L}C_{e}}$$
where *q_e_* is the amount of adsorbed substance by the weight unit of the solid; *Ce* is concentration of the adsorbed substance in the fluid phase in equilibrium condition with the solid matrix; *n_F_*, *e* and *K_F_* are constants added to the physical and chemical characteristics of the liquid phase and to the type of solid matrix; *q_MAX_* is the maximum adsorption capacity that is reached until the completion of the solid’s surface monolayer; and *K_L_* is defined as the Langmuir constant and is related to the energy.

## 3. Results and Discussions

### 3.1. Dye Abatement at Different pH

In the following Figure 1, it is possible to observe some representative examples of the dye abatement at different pH values. The first sample, subtitled “Basis”, represents the stock solution, which was stirred for 24 h without MWCNTs in order to demonstrate the absence of the abatement, which could be caused by the addition of the compounds used for changing the pH. On the other hand, the succeeding samples show the abatement due to the MWCNTs, which were used in the quantities defined previously and referred to stirring intervals 5 ÷ 240 min.

### 3.2. Transport Mechanisms

Adsorption takes place through a series of mechanisms that are difficult to characterise because they depend on many factors such as the affinity between solvent–solute, solid–solute or temperature. However, it was possible to differentiate the main phases of these mechanisms, which trigger an adsorbate mass transport from a liquid solution onto a solid adsorbent. The Weber and Morris method [48] was used, thanks to which it was possible to divide the experimental points into three areas: (i) transport within the solution with a molecular diffusive mechanism; (ii) transport within the liquid film present on the surface of the adsorbent solid matrix through intraparticle diffusion; and (iii) transport inside the pores of the adsorbent solid matrix. The liquid samples obtained from the batch tests were characterised using UV-VIS spectroscopy and the points obtained were represented by the Weber and Morris method in order to identify the different features of the three transport mechanisms. In Figure 2 and Figure 3, the experimental points for Reactive Blue 116 and Reactive Yellow 81 during the different pH analyses are shown. With the support of Table 1 and Table 2, it can be noted that, for both reactive dyes, the predominant transport mechanism is represented by the molecular mechanism, which occupies around 80% of the entire adsorption phenomenon, represented by the first zone. Subsequently, in the second zone, a short stretch with a lower slope identifies the intraparticle mechanism, while the third area, being sub-horizontal, can be neglected, therefore, it defines a lack of adsorption within carbon nanotubes pores. Hence, observing the arrangement of the experimental points at different pH values, it can be noted that the neutral pH represents a minimum point, while in a specular way, both the acid field and the base field have a faster transport. In fact, in the two models of RB-116 and RY-81, the maximum is reached for extreme pH (pH 2 and pH 12).

The Reactive Blue 116 is formed by a chelate complex having as its fulcrum the metal cation Cu^2+^ with coordination bonds Cu-N. The chelating effect at pH close to neutrality gave the complex a good stability, and the percentage of species formation varies according to the pH (Appendix A). In fact, alkalinisation determines the formation of hydroxides of the metal cation Cu^2+^, and consequently, the rupture of the complex, while the shift of the pH to acid values causes the protonation of the nitrogen atoms of the heterocyclic portions of the macrocycle, which are no longer available to establish coordination with metal. In this case, the concentration of the Cu^2+^ ion in solution increases significantly [51]. The Reactive Yellow 81, however, has a molecular structure characterised by a fair solubility at pH values close to neutrality, evidenced by the dissociation constant of the sulfonic groups. As much defined by their formation as they are defined by the relative stability of complex species, they are still closely dependent on the pH of the solution and the degree of protonation of the different nucleophilic sites present in the intact molecular structure of the dye and available for the coordination of the divalent species. With reference to the distribution diagram of complex species, it is observed that their molar fractions and their total number increase both at acidic and basic pH (Appendix A). The observed behaviour can be justified on the basis of the effect induced by the protonation/deprotonation equilibriums of the polar functions of the bond, since it cannot also exclude the contribution of the hydrolysis of the double bond N=N which would increase the number of potential free bonds in solution with high affinity for the metal cation. For both reagents, therefore, it is possible to hypothesise the intervention of the described phenomena of protonation/deprotonation and/or partial hydrolysis of the structures of the bonds used which, at acidic and basic pH values, favour and increase the adsorption kinetics of MWCNTs with respect to the neutrality conditions of the solution. All the experimental results obtained can therefore be interpreted on the basis of the above considerations.

### 3.3. Kinetic Studies

The adsorption kinetic study is very important because it can explain the link between the reaction rate, the reagents concentrations and the constant parameters [52]. Due to the reaction complexity [53], pseudo-order kinetic equations are used that also include in the constant parameters the compounds used to modify the pH, because they are not consumed by the reaction but, on the contrary, increase the velocity of the process. With the two equations models used (Equations (3) and (4)), it is imposed that the order of the kinetics is close to the values of the first and second order, allowing to neglect small correlation errors, proven by high *R^2^* values. The concentration used for both dyes (RB-116 and RY-81) was 100 mg L^−1^, while the nanotubes adsorbent solid matrix was always equal to 0.06 g. The time range for the experiments was between 5 ÷ 240 min, in order to evaluate the removal times. Figure 4 and Figure 5 represent the trends of the kinetic curves for the RB-116 and RY-81 in different pH analyses; they were referred to the pseudo-first order interpolation because it gave a better correlation, as shown in Table 3 and Table 4. For the RB-116, very fast adsorption kinetics were observed, especially for extreme pH (the curve at pH 12 reach the maximum *q_e_*), while a slight decrease was observed around neutrality. For the RY-81, kinetics faster than those of the Reactive Blue were observed, because its structure is characterised by a higher molecular weight, with the same weight concentration; there are fewer molecules in solution and they were immediately adsorbed on the solid matrix of nanotubes. In any case, the kinetics for both dyes were very fast and had the maximum at extreme pH. In the case of real textile waste, this result has a twofold significance; in fact, the drainage tanks have certainly non-neutral wastewater. Consequently, the waste removal process, which is carried out in situ, would be solved faster and with less costs than the normal processes.

### 3.4. Equilibrium Studies

The study of adsorption isotherms provided useful information to understand the variation of adsorption capacity among the concentration of solute in the fluid that has to be treated. The Freundlich and Langmuir models were used, because their results were closer to the transport mechanisms of Weber and Morris, therefore, they were closer to the pseudo-first order kinetics, which suggested diffusive molecular adsorption with the filling of all the active sites present on the adsorbent solid matrix surface. The analyses were carried out with batch stirring tests with constant time period (60 min) and at different dye concentrations (5, 30, 50, 100 and 150 mg L^−1^) and at different pH ranges (2, 4, 7, 9 and 12). Subsequently, the liquid sample analysed with the spectrophotometer returned the absorbance values with which we calculated the amount of solute adsorbed per unit of adsorbent, defined by and calculated using Equation (1), in which the initial concentration *C_0_* and the *C_e_* equilibrium (determined using UV-VIS analysis as absorbance) of the species in the solution were changed. The model with the best interpolation was given by the Langmuir equation; it is possible to observe that from Figure 6 and Figure 7 as well as from the analysis of the data in Table 5 and Table 6. Therefore, the reaction affected the formation of an adsorbate monolayer on the adsorbent solid surface until saturation.

Favourable isotherms were observed in both dyes, with specular trend respect to the neutrality field, which demonstrated the results obtained in the previous methods. In particular, for extreme pH, an almost straight curve was fitted, confirming the process speed, which did not allowed the formation of the characteristic horizontal asymptote, which represents the saturation of the surface layer.

## 4. Conclusions

Multi-walled carbon nanotubes were used successfully in this work as an adsorbent medium for the laboratory-replicated textile waste treatment. The sample batch tests allowed the identification of different experimental points, and their analyses characterised the transport mechanism, the kinetics and the isotherms. Specifically, the mass transport was analysed through the Weber and Morris method, concerning molecular diffusion phenomena for the dyes RB-116 and RY-81. In fact, interpreting the experimental points based on the three characteristic zones, a greater slope was fitted in the first section, where *q_e_* was reached in equilibrium equal to 30 mg_Dye_ g^−1^_CNTs_, which represented almost the 80% of the complete adsorption reaction. The residual percentage concerned the intraparticle diffusion mechanisms with a smaller slope, which required a longer time until the adsorbent solid matrix saturation. Once in equilibrium, small deviations were noticed due to the adsorption inside the pores that could be negligible. The HCl and NaOH compounds used for the modification of the pH led to the formation of different trends; in particular, at extreme pH (2 and 12), the maximum increase is fitted in the adsorption capacity as well as the process speed, while in the neutral field, the process was slower and less adsorption was achieved. The increase in efficiency for RB-116 in the acid field occurred because the metal was free in solution; consequently, the nitrogenous parts of the binder were subject to protonation equilibrium, while for the basic pH values, the formation of hydroxides and metal hydrates prevailed over the formation of the complex. On the other hand, for RY-81 both in the acidic and basic pH values, the double nitrogen–nitrogen bond was broken, which led to the formation of pseudo-aniline derivatives. Consequently, these dynamics determined only an increase in adsorption without the degradation of the dye, validating the experimental hypothesis.

The adsorption kinetics were described based on the better determination coefficient *R^2^* with the pseudo-first order model; this has shown that the process can be identified closer to the first order, but the equation is much more complex; in fact, compounds used (for the modification of the pH) were incorporated within the constant, since they are not consumed by the reaction, but, on the contrary, the speed of the adsorption process is increased. This proved once again, because the neutral pH value had more difficulties than extreme pH concentrations, in fact, that the last one created a much higher *q_e_* at equilibrium, equal to about 33 mg_Dye_ g^−1^_CNTs_.

As far as adsorption isotherms are concerned, there is a better interpolation of the experimental data through the Langmuir equation for both reagent dyes. This demonstrated and confirmed what has been extrapolated from the Weber and Morris methods, hence, the system developed a mass transport through a molecular diffusion mechanism that allowed the dye molecules to occupy the available active sites, which are positioned on the surface of the adsorbent matrix, forming a monolayer.

## Figures and Tables

**Figure 1 materials-13-02757-f001:**
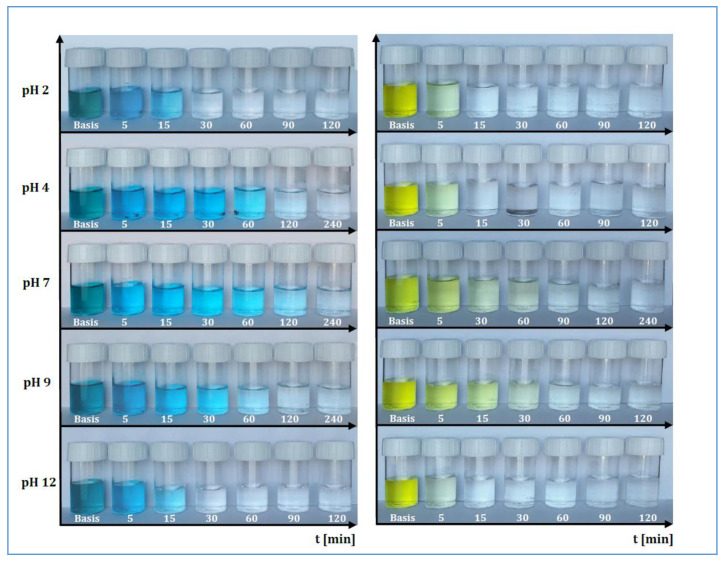
Temporal representation of abatement for RB-116 and RY-81 at different pH values.

**Figure 2 materials-13-02757-f002:**
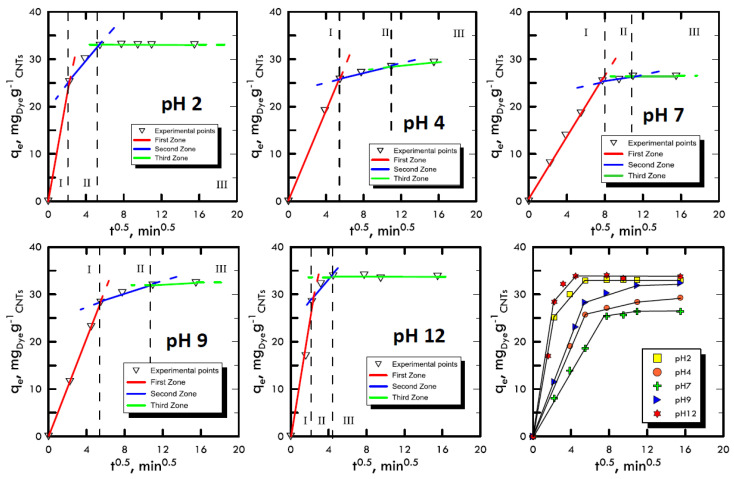
Weber and Morris method for the analysis of the transport mechanism RB-116 at different pH values.

**Figure 3 materials-13-02757-f003:**
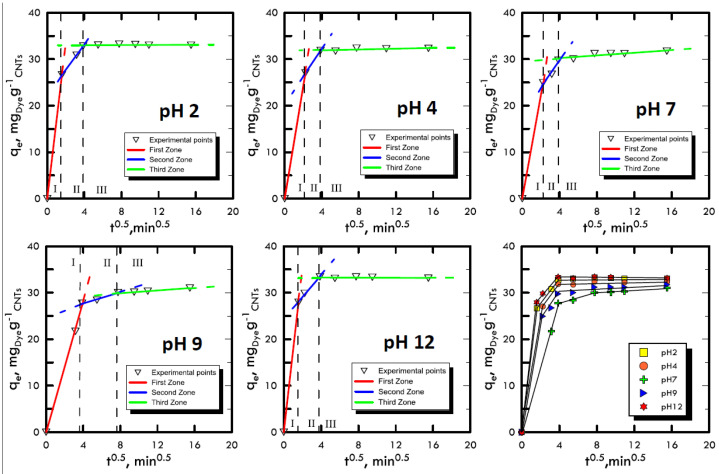
Weber and Morris method for the analysis of the transport mechanism RY-81 at different pH values.

**Figure 4 materials-13-02757-f004:**
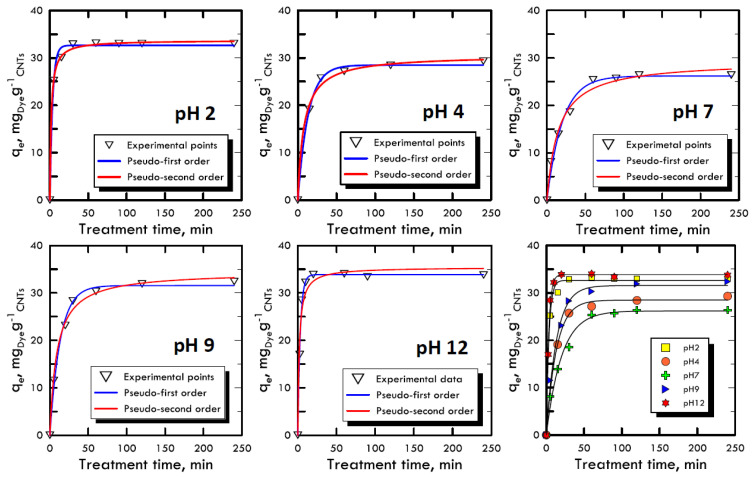
Kinetic adsorption model for Reactive Blue 116 at different pH values.

**Figure 5 materials-13-02757-f005:**
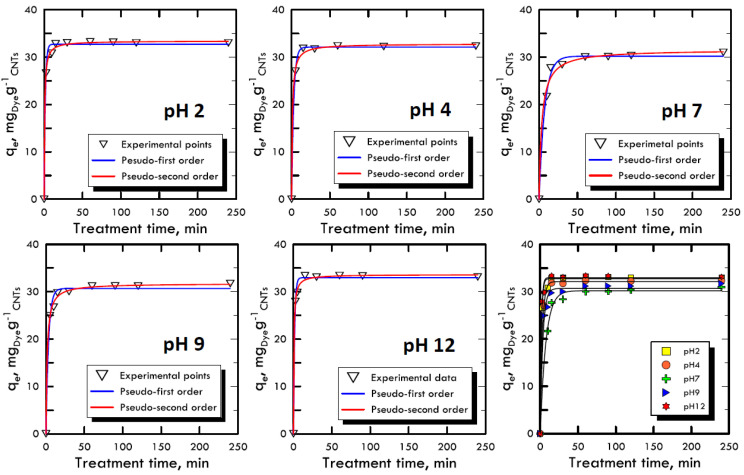
Kinetic adsorption model for Reactive Yellow 81 at different pH values.

**Figure 6 materials-13-02757-f006:**
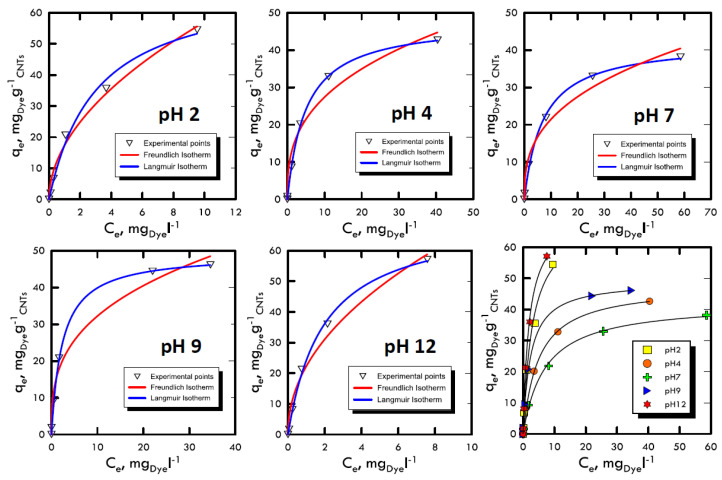
Adsorption isotherm for Reactive Blue 116 at different pH values.

**Figure 7 materials-13-02757-f007:**
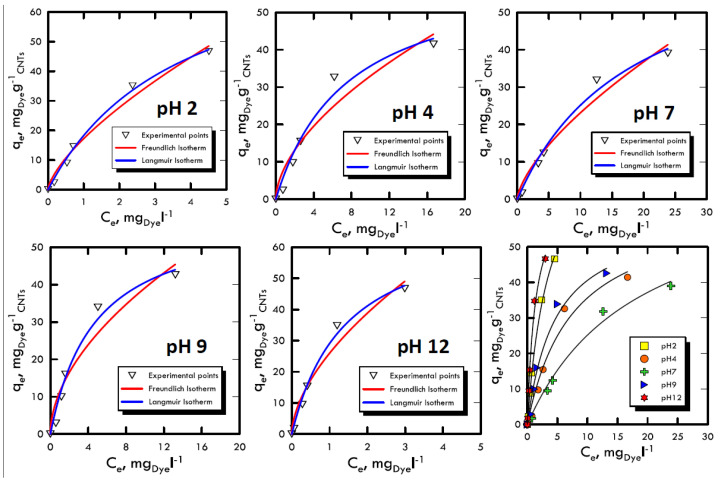
Adsorption isotherm for Reactive Yellow 81 at different pH values.

**Table 1 materials-13-02757-t001:** Return of Weber and Morris models for Reactive Blue 116.

BLUE 116	First Zone			Second Zone			Third Zone		
pH-Value	Slope	Angle	R^2^	Slope	Angle	R^2^	Slope	Angle	R^2^
2	11.26	84.92	1	2.37	67.12	0.997	0.003	0.17	0.945
4	4.72	78.03	0.998	0.48	25.64	0.981	0.198	11.19	1
7	3.27	72.99	0.997	0.30	16.69	0.925	4E-015	0	1
9	5.16	79.03	0.999	2.35	66.94	0.973	0.110	6.27	1
12	12.35	85.37	0.998	2.69	69.60	0.990	−0.014	−0.80	0.962

**Table 2 materials-13-02757-t002:** Return of Weber and Morris models for Reactive Yellow 81.

YEL-81	First Zone			Second Zone			Third Zone		
pH-Value	Slope	Angle	R^2^	Slope	Angle	R^2^	Slope	Angle	R^2^
2	16.84	86.60	1	2.69	69.60	0.999	0.007	0.40	0.995
4	12.08	85.26	1	2.95	71.27	1	0.040	2.29	0.939
7	7.06	81.93	0.997	0.60	30.96	0.998	0.148	8.41	0.999
9	11.14	84.87	1	2.90	70.97	0.995	0.141	8.02	0.973
12	17.67	86.76	1	2.33	66.77	0.999	−0.013	−0.74	0.941

**Table 3 materials-13-02757-t003:** Return of kinetic model data for Reactive Blue 116.

BLUE 116	Pseudo-First Order			Pseudo-Second Order		
pH-Value	k_1_	q_e_	R^2^	k_2_	q_e_	R^2^
2	0.2860	32.635	0.998	10.1390	0.0590	0.994
4	0.0745	28.474	0.997	1.9500	0.0654	0.994
7	0.0497	26.189	0.990	0.9877	0.0681	0.987
9	0.0740	31.547	0.994	1.9158	0.0580	0.993
12	0.3138	33.859	0.995	9.6900	0.0564	0.972

**Table 4 materials-13-02757-t004:** Return of kinetic model data for Reactive Yellow 81.

YELLOW 81	Pseudo-First Order			Pseudo-Second Order		
pH-Value	k_1_	q_e_	R^2^	k_2_	q_e_	R^2^
2	0.6685	32.734	0.998	26.269	0.0598	0.995
4	0.3678	32.128	0.999	16.760	0.0609	0.998
7	0.1367	30.203	0.999	4.503	0.0633	0.993
9	0.2980	30.682	0.997	10.976	0.0603	0.988
12	0.6970	32.968	0.998	32.010	0.0595	0.994

**Table 5 materials-13-02757-t005:** Return of adsorption isotherm data for Reactive Blue 116.

BLUE 116	Freundlich Isotherm			Langmuir Isotherm		
pH-Value	K_F_	1/n	R^2^	q_MAX_	K_L_	R^2^
2	17.370	0.516	0.988	70.106	0.331	0.993
4	12.196	0.351	0.970	47.743	0.203	0.999
7	9.386	0.329	0.978	42.238	0.142	0.997
9	15.081	0.350	0.983	49.201	0.426	0.998
12	22.306	0.478	0.980	71.054	0.518	0.998

**Table 6 materials-13-02757-t006:** Return of adsorption isotherm data for Reactive Yellow 81.

YELLOW 81	Freundlich Isotherm			Langmuir Isotherm		
pH-Value	K_F_	1/n	R^2^	q_MAX_	K_L_	R^2^
2	17.242	0.687	0.982	86.231	0.269	0.995
4	8.841	0.571	0.930	62.373	0.133	0.972
7	4.966	0.668	0.970	59.683	0.054	0.989
9	11.361	0.536	0.936	70.172	0.212	0.980
12	25.965	0.580	0.966	71.233	0.708	0.992

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
