# Peer review of "Adsorption of Reactive Blue 116 Dye and Reactive Yellow 81 Dye from Aqueous Solutions by Multi-Walled Carbon Nanotubes"

_materials, 2020, doi:10.3390/ma13122757_

Round 1

Reviewer 1 Report

The author investigated the adsorption of two dyes onto MWCNTs. The presented results might be used to develop water purification processes, but a number of questions need to be clarified before publication.

Although the Introduction cites 32 references, the authors must extend the description of the purpose of the work and its significance. Several cited works are not analyzed (see e.g. line 55 ref. 23-30). Furthermore, this section should be completed by the main aim of the work and the brief mention of the applied methods and/or conclusion, but the last paragraph (lines 60-69) of the present manuscript looks like a part of the Materials and Methods.

The origin of the MWCNT is not clear for me. Is it synthesized by the authors or received from the group of Ref 32? Noted here, several sentences are very similar to those written in Ref 32, even though they are two different research groups. Anyway, if the MWCNT synthesized by the authors, then the original paper must be cited. Furthermore, the relevance and properties of the produced materials be must investigated and the related data and spectra must be inserted into Supplementary Materials. In case the MWCNT has been received from a group or bought from a supplier, this should be clearly described by indicating the source of the sample. Furthermore, the known properties of the materials (purity, diameter, length, specific surface area, etc.) must be given too.

Repetitions are found, e.g. line 78-85 is similar to line 95-105. Please clarify it.

“the wastewater is filtered with the aid of a vacuum pump to allow the separation of the liquid phase from the solid phase” But how? The properties of the filter must be inserted. It is well-known that the presence of MWCNT in the solution influences the measured absorption spectra.

Figure 1 and its discussion must be transferred to the Results and Discussion.

“triple reading of the samples” means that the same samples have been measured three times? Because, in my opinion, at least 3 samples prepared independently should be measured. Furthermore, some statistics also must be inserted (the presented data are the average of the independent measurements? their standard deviation? etc)

HCl and NaOH were used for setting the pH during the sample preparation. As the solvent has not any buffering capacity, the pH will change during the adsorption process. How was changed the pH? The related data must be inserted and analyzed. How was the pH of the calibration curve determined for each sample?

In my opinion, the results can only be interpreted after the sample preparation has been clarified and some basic information about the properties of the MWCNTs has been given.

The authors write wastewater everywhere despite working in a model solution. How do other molecules and ions present in the actual wastewater affect the investigated adsorption process?

A few more notes:

Abbreviations are inappropriate. The abbreviation of the words must be given in parentheses after the first occurrence of the given word. Subsequently, the abbreviated form must be used for the given word. I do not suggest using any abbreviations in the abstract.

Many of mistypes can be found in the manuscript.

Therefore, I suggest reconsidering the publication after major revision or resubmission.

Author Response

Dear Reviewer,

thank you for your time spent reviewing our manuscript and for giving us important tips for improving it. The manuscript has been revised in many parts. The changes have been highlighted in red. The attached file contains all the answers point by point.  We hope that in this form the manuscript will find your approval to be considered for its publication.

We thank you and send you our best regards.

Reviewer 2 Report

This manuscript investigates the adsorption of two dyes from water using the self-prepared carbon nanotubes. Although the research in this field is already a crowded one, this manuscript presents interesting results. I appreciate the data and analysis of the pH effect. However, extensive editing of English language and style/format is required.

The major issue of this manuscript is the language problem. Authors are recommended to send the manuscript to an English editor for language check and revision.

Major issues

  1. Although pseudo-first and -second-order rate equations have been widely used for adsorption kinetic data fittings, the misapplications of those equations have been found (AIChE Journal, 64(5) 2018, 1793, DOI 10.1002/aic.16051). I suggest the authors add a short discussion about this issue by citing the highlighted paper.
  2. The first two paragraphs in the introduction (lines 33 to 42) are actually common sense to the researchers. More specific contents relating to the work in the present study should be included. Using carbon nanotubes as adsorbent is not new in the field. The state-of-art of the related research is recommended to added.

Minor issues

  1. Figure 1 seems to be the results. I guess it may be better to put it in the results and discussion section.
  2. The expression of some units does not look professional. See line 144 and 145. Please check the whole manuscript and use ISO unit for all parameters.

Author Response

Dear Reviewer,

thank you for your time spent reviewing our manuscript and for giving us important tips for improving it. We have really appreciated your kind and delicate ways to suggest your comments. The manuscript has been revised in many parts. The changes have been highlighted in red. The attached file contains all the answers point by point.  We hope that in this form the manuscript will find your approval to be considered for its publication.

We thank you and send you our best regards.

Round 2

Reviewer 1 Report

Despite the authors made effort to answer the questions about sample preparation, in my opinion, the description of the relevant part is still inaccurate, or the sample preparation was not adequate.

Line 110-114: “During the adsorption process, the system was always subjected to mechanical stirring and the pH was constantly monitored by the use of a pH meter previously calibrated with the appropriate standards. It was kept constant by adding acid or basic solution drops as needed. Buffer solutions have not been deliberately used to avoid the presence of other compounds in the system.” This means, that the initial pH of the dye solution was kept constant by the continuous addition of acid and/or basic solution during the adsorption process until the separation. Therefore, the Na+ and Cl- content of the samples were different from the calibration solutions. How was taking into account the effect of the presence of these ions on the absorption spectra of the investigated dyes? Furthermore, how was considered the effect of the continuous addition of these ions on the adsorption process?

Line 121-122: “Once the stirring time was over, carbon nanotubes, after the adsorption process, were separated from the dye solution by a vacuum pump.” Vacuum pump is not a separation process. Was evaporation, distillation or filtration used? In the case of the first two, a number of questions arise. In the case of filtration, the properties of the filter must be inserted, and the adsorption of the dyes must be investigated onto the filter. This is one of the important points of sample preparation because the separation process can affect the concentration of the dyes (dyes adsorption onto the filter) and the absorption spectra (presence of CNTs, presence of the different ions).

In my opinion, the results can only be interpreted after the sample preparation has been clarified.

Author Response

Dear Reviewer,

we have taken all your suggestions into consideration. Actually, we believe that it was necessary to better explain some phases of our work. We hope that in this version it will find your approval.

We thank you for the patience you have shown to us and for the time spent. The changes made since the last version are listed below.

With best regards

1)Line 110-114: “During the adsorption process, the system was always subjected to mechanical stirring and the pH was constantly monitored by the use of a pH meter previously calibrated with the appropriate standards. It was kept constant by adding acid or basic solution drops as needed. Buffer solutions have not been deliberately used to avoid the presence of other compounds in the system.” This means, that the initial pH of the dye solution was kept constant by the continuous addition of acid and/or basic solution during the adsorption process until the separation. Therefore, the Na+ and Cl- content of the samples were different from the calibration solutions. How was taking into account the effect of the presence of these ions on the absorption spectra of the investigated dyes? Furthermore, how was considered the effect of the continuous addition of these ions on the adsorption process?

1) Answers. For clarity the following sentences have been added to the manuscript:  “The pH changes with respect to the initial system, during the adsorption process, were minimal, with a variation of pH= + 0.1 such as to require sporadic additions of a few drops of diluted acid or base solution (0.01M). Therefore the variation in Na+ and / or Cl- content was considered completely negligible. Moreover, since the volumes of acid or base added to adjust the pH of very little have been very small, it can be assumed that the overall ionic strength of the final solution varies by a completely negligible amount compared to that of the initial solution, thus not influencing the process of adsorption(lines 114-120).

For each UV analysis, "white" reference solutions were used, not containing dye but the same quantities of acid or added base”(lines 134-135).

2) Line 121-122: “Once the stirring time was over, carbon nanotubes, after the adsorption process, were separated from the dye solution by a vacuum pump.” Vacuum pump is not a separation process. Was evaporation, distillation or filtration used? In the case of the first two, a number of questions arise. In the case of filtration, the properties of the filter must be inserted, and the adsorption of the dyes must be investigated onto the filter. This is one of the important points of sample preparation because the separation process can affect the concentration of the dyes (dyes adsorption onto the filter) and the absorption spectra (presence of CNTs, presence of the different ions).

2) Answer. The following sentence has been added to the manuscript for greater clarity: “Once the stirring time had elapsed, after the adsorption process, the carbon nanotubes were separated from the solution by filtration through a vacuum pump. The filter used was .a borosilicate glass filter funnel  (Pyrex- porosity 5). At the end of the separation process the filter remained colorless allowing to exclude an adsorption of the dye by the filter. The solution obtained was clear without any presence of nanotubes.” (lines 127-131.)

Reviewer 2 Report

The current version can be accepted for publication.

Author Response

We thank  the reviewer for the time spent and his valuable suggestions
With best regards

Round 3

Reviewer 1 Report

The authors made effort to improve the description of the materials and methods. The results are fine, I suggest checking the English language.